# A Spectroscopic Study of Solid-Phase Chitosan/Cyclodextrin-Based Electrospun Fibers

## Chen Xue and Lee D. Wilson *

Department of Chemistry, University of Saskatchewan, 110 Science Place, Saskatoon, SK S7N 5C9, Canada; chx257@mail.usask.ca

\* Correspondence: lee.wilson@usask.ca; Tel.: +1-306-966-2961; Fax: +1-306-966-4730

**Abstract:** In this study, chitosan (chi)/hydroxypropyl-β-cyclodextrin (HPCD) 2:20 and 2:50 Chi:HPCD fibers were assembled via an electrospinning process that contained a mixture of chitosan and HPCD with trifluoroacetic acid (TFA) as a solvent. Complementary thermal analysis (thermal gravimetric analysis (TGA)/differential scanning calorimetry (DSC)) and spectroscopic methods (Raman/IR/NMR) were used to evaluate the structure and composition of the fiber assemblies. This study highlights the multifunctional role of TFA as a solvent, proton donor and electrostatically bound pendant group to chitosan, where the formation of a ternary complex occurs via supramolecular host–guest interactions. This work contributes further insight on the formation and stability of such ternary (chitosan + HPCD + solvent) electrospun fibers and their potential utility as "smart" fiber coatings for advanced applications.

**Keywords:** cyclodextrin; chitosan; electrospinning; fiber; assembly

## 1. Introduction

Chitosan is a copolymer containing β-(1-4)-linked N-acetyl-D-glucosamine and D-glucosamine monomer units derived from the deacetylation of chitin, where the degree of acetylation depends on the extent of hydrolysis [1–7]. The amino polysaccharide units of chitosan contribute to its biodegradability and biocompatibility, along with its unique physicochemical properties relevant to adsorption. According to the degree of ionization of glucosamine groups of chitosan, the antimicrobial and adsorption properties can be altered by variable pH levels [8–11]. Diverse applications of chitosan have been found in many fields that exploit its pH-dependent binding properties, as evidenced in wastewater treatment, food preservation and biomedical devices [8,9,12–18]. To attain additional performance in these applications, research has also focused on the design of different morphological forms of chitosan that include nanofibrous systems, owing to the high surface area of these biomaterials [2–4,14,16–20]. Among the various approaches in the design of fibrous and nanofibrous materials, electrospinning methods have gained increasing attention [16–18,21–24]. While chitosan is soluble in acidic aqueous solution, technical difficulties related to the electrospinning of uniform fibers arise due to repulsive interactions between the cationic sub-units of chitosan that attenuate chain entanglement effects [4,19–21,25]. The judicious choice of additive components offers a solution to offset such charge repulsion effects during electrospinning [4,7,26–30]. For example, Burns et al. reported the use of trifluoroacetic acid (TFA) and hydroxypropyl β-cyclodextrin (HPCD) as additives to assist in the electrospinning of chitosan nanofibers [31]. The potential of HPCD to form noncovalent host–guest complexes with various molecular species, in conjunction with the polyelectrolyte nature of chitosan, may further enhance the utility of such nanofibrous materials as advanced coating materials [31,32]. Notwithstanding the complexation properties of HPCD, the molecular level details that account for the uniform formation of chitosan fibers via this pathway are not well understood. While the authors

allude to the possibility that inclusion complex formation occurs between chitosan and HPCD, further insight is required to establish the origins of the improvement of fiber formation in this ternary (HPCD + chitosan + TFA) system to advance the field of chitosan nanofiber materials.

Herein, the overall goal of this study relates to an investigation of the structural role of ternary components (HPCD + chitosan + TFA) in the solid state via complementary spectroscopic techniques for chitosan-based fibers. The results of this study are foreseen to provide insight on the fiber formation process in such ternary component systems that will aide in the development of improved electrospinning formulations for chitosan systems. To address this goal, preparations of Chi:HPCD fibers at variable ratios were carried out via electrospinning in nonaqueous media. The composition and structural characterization of the electrospun fibers in the solid state was carried out using thermal analysis and spectroscopic (FT-IR and Raman) methods. Raman spectral imaging was assisted by a rhodamine dye probe to gain further structural information on the chitosan fiber composite materials reported herein.

## 2. Materials and Methods

### 2.1. Materials

Research grade 6-O-hydroxypropyl β-cyclodextrin (HPCD) (degree of substitution (DS) = 4.6), was purchased from CYCLOLAB, Ltd. (Budapest, Hungary) and was used as received. Rhodamine 6G was purchased from Allied Chemical (Morristown, New Jersey, NJ, USA) and used as received. Low molecular weight (LMW) chitosan (Chi, 75–80% deacetylation and molecular weights in a range of 50–190 kDa measured by Brookfield viscosity 20 cps), deuterated dimethyl sulfoxide (DMSO-$d_6$, 99.9%), tetrahydrofuran (THF) and trifluoroacetic acid (TFA) was bought from Sigma-Aldrich Canada Ltd. (Oakville, ON, Canada).

### 2.2. Solution Preparation

All solutions used for electrospinning were formulated according to Burns et al. [31], where the designated mass of chitosan and HPCD (wt%) was added to neat TFA and allowed to mix overnight with stirring (100 rpm) at 23 °C. All solutions were kept at 4 °C and consumed within 72 h.

### 2.3. Electrospinning

Solutions prepared in Section 2.2 (2.5 mL) were placed into a 10 mL syringe with a metal needle (Inner Diameter = 0.508 mm) that was operated by a Cole-Parmer 78-0100c syringe pump (Cole-Parmer, Montreal, QC, Canada). The distance between the needle tip and collector plate was set to 11.5 cm. The flow rate was controlled at 0.1 mL h$^{-1}$, where a high voltage was produced by a high voltage power supply (Spellman CZE 1000R) (Spellman, Hauppauge, NY, USA) and was applied between the needle tip and collector plate during electrospinning. The voltage was slowly adjusted from 7 to 15 kV to achieve a stable Taylor cone. The resulting fiber product was accumulated onto a foil-covered, electrically grounded collector plate. The electrospinning apparatus and molecular structures of precursors and solvent is shown in Scheme 1.

### 2.4. $^1H$ NMR Spectroscopy in Solution

The HPCD content of the fiber samples was estimated using a quantitative NMR (qNMR) method that was adapted from a previous report [33]. A 1% (w/w) THF/DMSO-$d_6$ solution was prepared by adding a desired amount of THF to DMSO-$d_6$ to which ca. 5 mg of the *as-spun* fiber was dissolved with ~600 mg of solvent (THF/DMSO-$d_6$) in a 5 mm NMR tube. $^1H$ NMR spectra were obtained using a wide-bore (89 mm) 11.7T (500 MHz) Oxford superconducting magnet system (Bruker BioSpin Corp., Billerica, MA, USA) equipped with a 5 mm Pa Tx1 probe. THF served as an internal quantitative standard for estimation of HPCD content of a fiber sample.

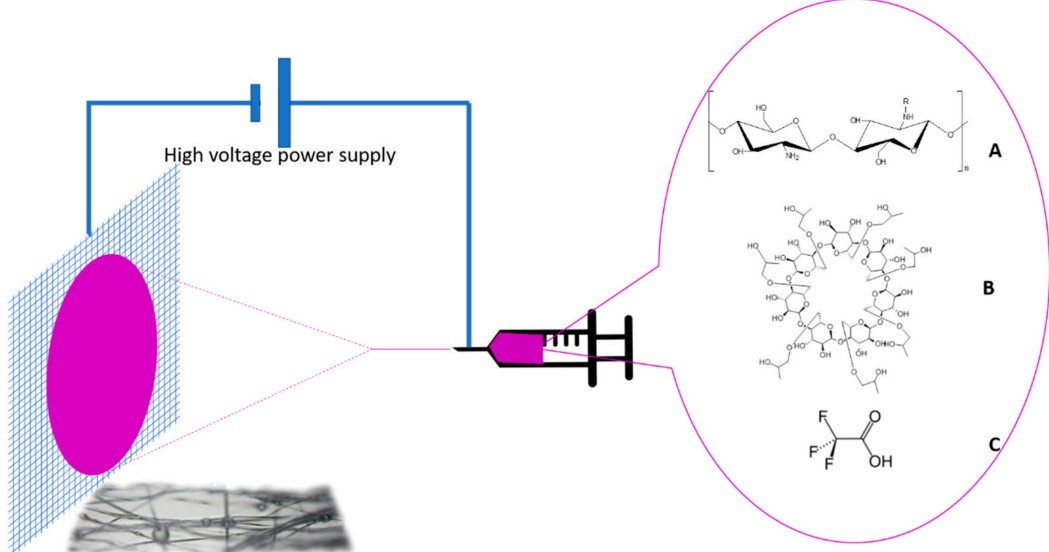

**Scheme 1.** A schematic illustration of the electrospinning setup and molecular structure of the precursors: (**A**) chitosan where R = –COCH$_3$ or H depending the degree of deacetylation, where n depends on the relative molecular weight of the biopolymer; (**B**) hydroxypropyl-β-cyclodextrin (HPCD); (**C**) trifluoroacetic acid (TFA) is the solvent system.

### 2.5. FT-IR Spectroscopy

Reflectance-based IR spectra were obtained with a Bio-RAD FTS-40 (Bio-RAD Laboratories, Inc., Hercules, CA, USA) instrument. IR samples were prepared by thoroughly mixing a sample (~5 mg) with FT-IR grade KBr (~50 mg) with a mortar and pestle. FT-IR spectra were measured at 23 °C with a resolution of 4 cm$^{-1}$ over a spectral range of 400–4000 cm$^{-1}$ (Kubelka-Munk intensity units) where the background spectrum of KBr was subtracted. Multiple scans (n = 16) were collected for better signal-to-noise ratio.

### 2.6. Raman Spectroscopy

A Renishaw InVia Reflex Raman microscope (785 nm solid state diode laser with a 1200 lines/mm grating system) (Renishaw plc, New Mills, UK) was used to collect One-dimensional (1–D) Raman spectra with a Pelletier cooled CCD (charge coupled device) detector (400 × 576 pixels). The instrument wavelength was calibrated at 520 cm$^{-1}$ using an internal Si (110) sample.

### 2.7. Raman Imaging with Dye Probe

A procedure was adapted from a previous reported work for acquiring Raman spectra [34]. High resolution Raman 2-D (two-dimensional) spectral imaging (1.1 × 1.1 μm pixel size) was acquired using the instrument described above with a Leica 50× long-working-distance objective (numerical aperture = 0.50) using Streamline mode within the instrument software (Renishaw Wire V3.4) (Renishaw plc, New Mills, UK) with a 12 s exposure time under a static scan mode centered at 790 cm$^{-1}$. An effective spectral range of 596–985 cm$^{-1}$ is achieved by the monochromator grating. 2-D images (92 × 70 spectra) were collected at specific Raman shifts in a uniform 101 × 77 μm grid to create Raman micro-images with spectral intensity (relative to baseline) for respective Raman shifts. Prior to creation of Raman images and to minimize any baseline sloping and offset effects that result from experimental artefacts, we performed the baseline linearization and normalization of the spectra for Raman imaging using the built in Renishaw Wire V3.4 method. The color intensity of the pixels on the image corresponded to the integration of the respective bands (610 and 850 cm$^{-1}$) to baseline as required. To highlight the chitosan spectral region, the results from dividing the band integrations at 610 cm$^{-1}$ by the band at 850 cm$^{-1}$ were used to construct the Raman images for chitosan. Prior to the Raman imaging, dried Chi:HPCD

2:50 sample was placed onto an Au-coated Si wafer that was then soaked with a Rhodamine 6G dye in a benzene solution (~0.1 µM), and finally dried under ambient ventilation for 24 h.

### 2.8. Differential Scanning Calorimetry (DSC)

A TA Q20 thermal analyzer (TA instruments, New Castle, DE, USA) was used to obtain DSC profiles of HPCD, physical mixtures of components, chitosan and 2%:50% fiber over the range between 40 °C and 190 °C. Solid samples were hermetically sealed in aluminum pans, where the sample weight varied from 1.55 mg to 2.60 mg before spectral acquisition. DSC profiles were recorded at a 10 °C/min scan rate under dry nitrogen gas with a flow rate at 50 mL/min.

### 2.9. Thermal Gravimetric Analysis (TGA)

A Q50 (TA instruments, New Castle, DE, USA) thermogravimetric analyzer was employed to obtain weight loss profiles. The experiments were executed under a nitrogen atmosphere with a heating rate (5 °C min$^{-1}$) up to a maximum temperature (500 °C). First derivative plots (weight %/°C vs. temperature (°C)) were generated to determine the thermal stability of materials.

### 2.10. Scanning Electron Microscopy (SEM)

Scanning electron microscopy (SEM) images with 508 dpi resolution were acquired with a JSM-6010LV microscope (JEOL, Ltd., Tokyo, Japan) at various magnifications (1000×, 5000× and 3000×). The samples were fixed onto a sample mounting stub with conductive carbon tape.

## 3. Results and Discussion

As indicated in the introduction section above, the role of additives (TFA, HPCD) for the assisted electrospinning of chitosan are not well known. Therefore, structural characterization of the fiber products was carried out to further understand the structure and composition of these composite materials, as described in further detail below.

### 3.1. SEM Results

SEM images of Chi:HPCD fiber are shown in Figure 1. We observed a mixture of products that possess nodule-shaped elements and fiber-like morphology in both Figure 1A,B that show the resulting materials for both ratios (2:20 and 2:50) with a heterogeneous composition. It was noted that the Chi:HPCD 2:50 fiber contained less nodule-shape elements and had a large fiber diameter when compared with the Chi:HPCD 2:20 fiber. These general observations coincide with those reported in the previous study by Burns et al. [31].

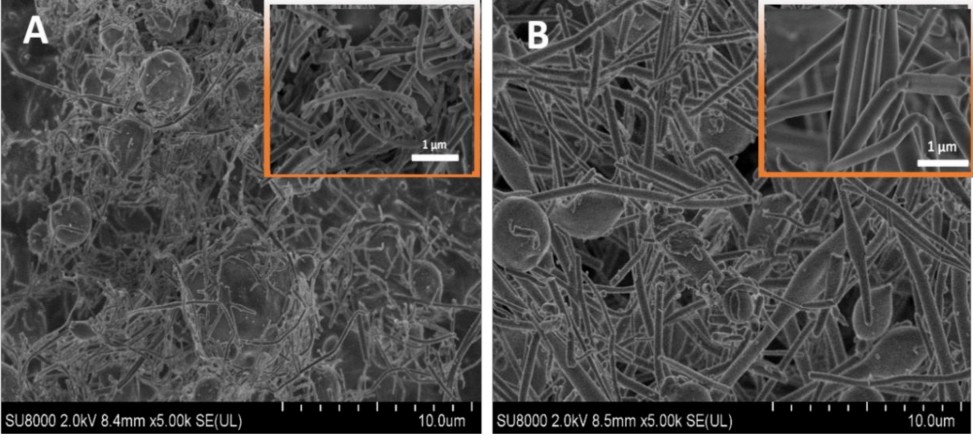

**Figure 1.** Scanning Electron Microscopy (SEM) images of chitosan (chi)/hydroxypropyl-β-cyclodextrin (Chi:HPCD) fibers. (**A**) Chi:HPCD 2:20 and (**B**) Chi:HPCD 2:50.

### 3.2. Determination of HPCD Content in the As-Spun Fibers

To assess the composition of the Chi:HPCD electrospun fibers, the HPCD content in the *as-spun* fibers was examined using $^1$H NMR spectroscopy. The composition of HPCD (wt. %) in the *as-spun* fibers was calculated based on the integration of a –CH$_3$ group of HPCD ($\delta$ = 0.99 ppm) relative to the internal standard THF ($\delta$ = 1.60 ppm). The $^1$H NMR spectra are shown in the accompanying Supplementary Materials. To verify the accuracy of the method, a quantitative determination of a known amount of HPCD was evaluated in a test sampling trial. The results are summarized in Table 1, where the measured content (wt. %) of HPCD in the *as-spun* fiber sample was drastically lower than that of the theoretical value (the composition according to mass ratios in the prepared solution). The Chi:HPCD 2:20 fiber had ~50% difference and the Chi:HPCD 2:50 fiber had ~20% difference, as compared with the respective theoretical values based on wt. % content. This suggests that solvent effects may contribute to the weight of the *as-spun* fiber, which yields a reduced composition (wt. %) of HPCD. This difference also infers that there was more solvent residue in the 2:20 fiber than the 2:50 fiber.

**Table 1.** Determination of hydroxypropyl-$\beta$-cyclodextrin (HPCD) content in the *as-spun* Chi:HPCD fiber using $^1$H NMR spectroscopy.

| Material | Determined HPCD Content | Theoretical Value * |
|---|---|---|
| HPCD | 100% (4.47 mg) [1] | 100 ± 6.9% (4.80 mg) [1] |
| Chi:HPCD 2:20 Fiber | ~40% | ~91% |
| Chi:HPCD 2:50 Fiber | ~75% | ~96% |

\* Theoretical value was calculated based on mass ratios between chitosan and HPCD in the prepared solution, assuming that all the solvent was evaporated from fibers. [1] Values in parentheses refer to the actual sample weight used for the qNMR calibration (cf. Section 2.4).

### 3.3. FT-IR Results of As-Spun Fibers

The Chi:HPCD fibers were further characterized by FT-IR to identify the composition of the *as-spun* fiber. The FT-IR spectra of *as-spun* fibers are presented in Figure 2 without normalization. FT-IR spectrum of pristine chitosan was shown in Supplementary Materials (Figure S1A). The IR band at 1786 cm$^{-1}$ relates to a free –COOH group in TFA [35–38]. The intensity of this band for Chi:HPCD 2:20 was higher than the value estimated for Chi:HPCD 2:50. This further indicates that there is higher TFA content in the 2:20 fiber than the 2:50 fiber system, which agrees with the results in Table 1. The –COO$^-$ band at 1679 cm$^{-1}$ corresponded to that of the trifluoroacetate ion [35]; whereas the band at 1526 cm$^{-1}$ was assigned to the protonated amine (–NH$_3^+$) group of chitosan [34,39,40]. These signatures indicate that electrostatic interactions are likely to occur between the –COO$^-$ group from TFA with the cation sites (–NH$_3^+$) of chitosan within the fiber composite. The band at 1221 cm$^{-1}$ and the shoulder at 1183 cm$^{-1}$ was assigned to the vibrational signature of C–F for TFA [36–38]. The two spectral signatures disappear in the 2:50 fiber, but were evident for the 2:20 fiber. The spectral variation between samples is the result of interference of free TFA in the 2:20 fiber, as it contains more TFA over the 2:50 fiber, according to the qNMR results in Section 3.2 and the IR band intensity results of the free –COOH group (1786 cm$^{-1}$) of TFA [35–38]. Hence, the vanishing of two signatures at 1221 and 1183 cm$^{-1}$ in the 2:50 fiber can infer that the –CF$_3$ group of TFA is bound by the annular region of the HPCD host to form a stable complex. An analogous inclusion mode for $\beta$-cyclodextrin-guest complexes that were formed between volatile organics with a trifluoromethyl group have been reported by Wilson and Verrall [32] for a range of $\beta$-CD/halothane systems (*cf*. Scheme 2 in [32]).

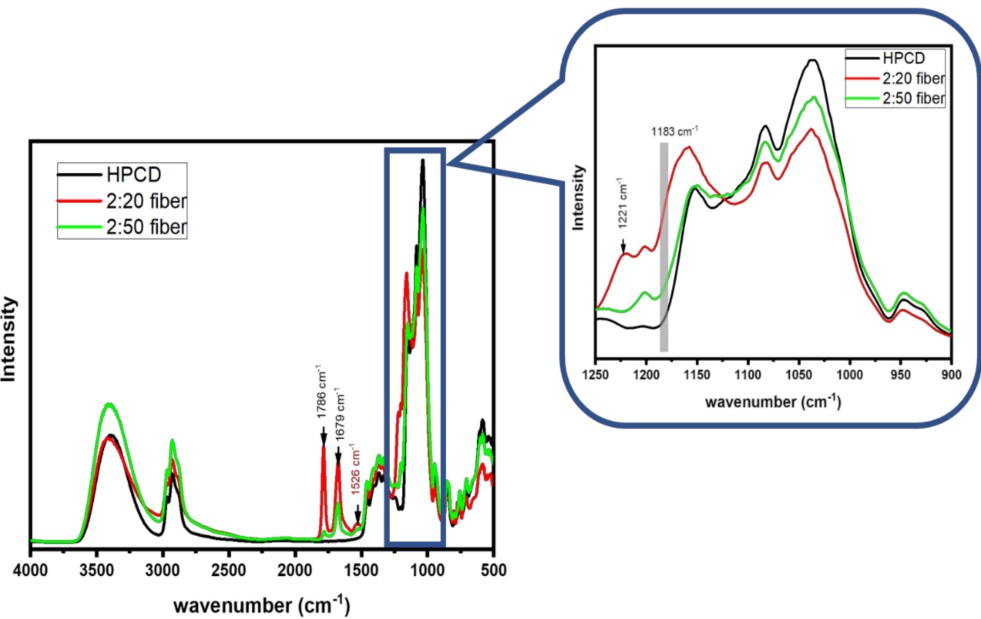

**Figure 2.** FT-IR spectra of HPCD and *as-spun* Chi:HPCD fiber (2:20 and 2:50, respectively) without normalization. The expanded region of the inset spectra is shown between 1250 and 900 cm$^{-1}$ of the original spectra.

### 3.4. TGA and DSC Results of Chi:HPCD Fiber

The *as-spun* Chi:HPCD 2:50 fiber was characterized by TGA and DSC, where the 2:20 *as-spun* fiber was not measured because it contained higher free TFA, which was done in order to avoid potential damage to the instrument (DTG plot of chitosan was given in Supplementary Materials, Figure S1B). Differential thermogravimetric (DTG) plots of HPCD and Chi:HPCD 2:50 are shown in Figure 3A, where a thermal event for HPCD at ~350 °C showed no apparent temperature shift, as compared with that of the fiber prepared at the 2:50 ratio. The 2:50 fiber displayed a new thermal event at ~260 °C that corresponds to the decomposition of a trifluoroacetate salt [41]. This indicates that the major fraction of TFA in 2:50 fiber is in the form of a trifluoroacetate salt ($CF_3COO^-$/$-NH_3^+$), where the $-NH_3^+$ groups are the protonated glucosamine sites of chitosan. This trend is consistent with the FT-IR results reported herein (*cf.* Figure 2). According to the DSC results in Figure 3, evidence of mixing occurs between chitosan and HPCD. However, because of the greater mass content of HPCD in the binary system (chitosan + HPCD), the thermal analysis (TGA and DSC) results for the Chi:HPCD fiber may be obscured, precluding further detailed compositional analysis of the fiber material.

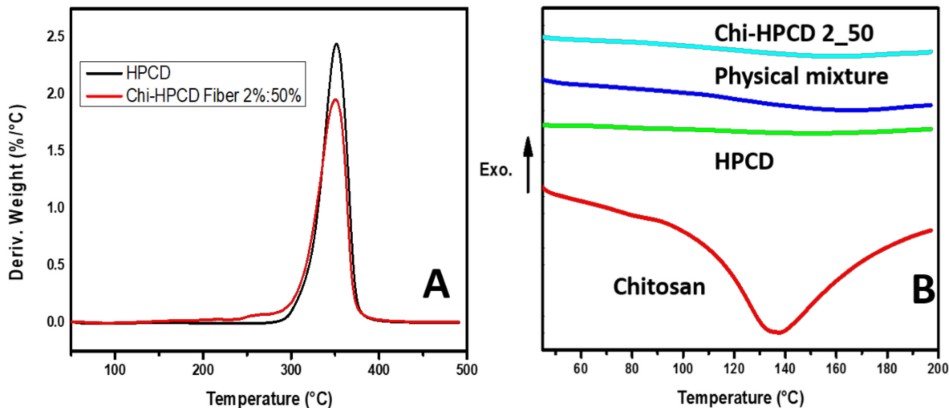

**Figure 3.** Thermal analysis results of *as-spun* Chi:HPCD fiber. (**A**) DTG plots of HPCD and Chi:HPCD 2:50 fiber; and (**B**) DSC profiles of chitosan, HPCD, physical mixture, and *as-spun* Chi:HPCD 2:50.

### 3.5. Raman Results

The Raman spectra of HPCD, pristine chitosan, *as-spun* Chi:HPCD fibers and Chi:HPCD fibers after 3 days and 3 months are demonstrated in Figure 4. The band at 730 cm$^{-1}$ and 1788 cm$^{-1}$ was assigned to the deformation of –COO$^-$ and the stretching of carbonyl group in trifluoroacetate anion, respectively [42]. The relative intensity of this band to other spectral signatures revealed a gradual decrease between the *as-spun* fiber (Figure 4C) to an aged fiber after 3 days (Figure 4E), then to a further aged fiber after 3 months (Figure 4F). This demonstrates that TFA content in the fiber decreases slowly over time. A noticeable fact is that signatures related to the –CF$_3$ group (1143, 1202, 601 and 521 cm$^{-1}$) [42] were not clearly observed in the Raman spectra, in contrast to the –COO$^-$ group of TFA, which suggests that the –CF$_3$ group may form an inclusion complex with HPCD. This conclusion is supported by the FT-IR results, as evidenced by the disappearance of bands at 1221 and 1183 cm$^{-1}$ in Figure 2. Expansions of the original spectra in the range of 770 and 990 cm$^{-1}$ are similarly shown in Figure 4. The Raman signatures over this spectral range relate to the variation of the C–O–C torsional angle, since it is sensitive to its local conformation [34,43]. For HPCD, the band at 850 and 925 cm$^{-1}$ was related to ring breathing of HPCD, while the band at 948 cm$^{-1}$ related to the skeletal vibration for the α-1,4 linkage of HPCD [44]. A comparison was made among precursors (ie, HPCD and chitosan) and different Chi:HPCD fiber types. New bands were observed at 827, 864, 884 and 918 cm$^{-1}$. By comparison, the band at 950 cm$^{-1}$, when compared with the spectrum of HPCD in Figure 4A, displayed minimum change. Among the new spectral bands observed, none were associated with the signatures of the CF$_3$COO$^-$ anion when compared with the results reported by Robinson and Taylor's Raman spectral study of the trifluoroacetate ion [42]. These bands showed maximum relative intensity for the *as-spun* Chi:HPCD 2:50 fiber spectrum (Figure 4C) and minimum relative intensity in the spectrum of Chi:HPCD 2:50 fiber after 3 months (Figure 4F). Therefore, new spectral bands at 827, 864, 884 and 918 cm$^{-1}$ may be related to conformational changes of the glucopyranose unit of HPCD that result from the interfacial host–guest complex [32] formed between the –CF$_3$ group of TFA and HPCD. The attenuation of the relative intensity of these spectral bands may be ascribed to attenuated formation of such complexes between TFA and HPCD, as the trifluoroacetate ion content of the fiber decreased. Similar Raman spectral changes have been reported elsewhere [45,46]. Furthermore, the spectral band shapes centered at 850 and 925 cm$^{-1}$ for the Chi:HPCD 2:50 fiber after 3 months in the spectrum (Figure 4F) was similar to the HPCD precursor (Figure 4A), but revealed broader Raman spectral features. By comparison, the band at 948 cm$^{-1}$ showed little difference between HPCD (Figure 4A) and the Chi:HPCD fibers (Figure 4C–F). This effect may suggest that while the –CF$_3$ group of TFA was bound to the interfacial region of HPCD, there was limited host–guest interaction with the α-1,4 linkage domain of HPCD. As well as this, the compositional difference between chitosan and HPCD precluded a diagnostic spectral analysis to assess the details of the host–guest interaction between chitosan and HPCD due to the presence of excess HPCD in the system.

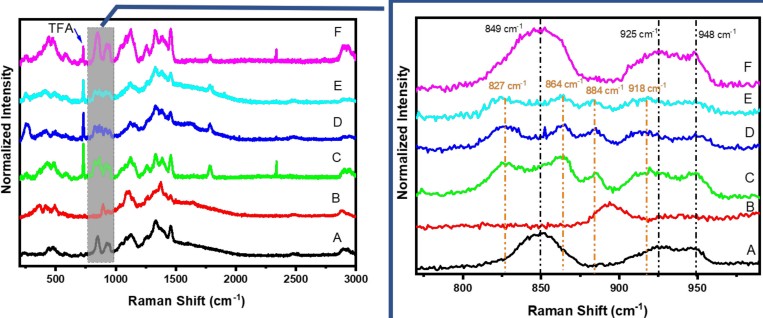

**Figure 4.** Raman spectra of precursors and different Chi:HPCD fibers: (**A**) HPCD, (**B**) pristine chitosan, (**C**) *as-spun* Chi:HPCD 2:50 fiber, (**D**) *as-spun* Chi:HPCD 2:20 fiber, (**E**) Chi:HPCD 2:50 fiber after 3 days and (**F**) Chi:HPCD 2:50 fiber after 3 months. The expanded region (**Right**) cover the region between 770 and 990 cm$^{-1}$.

### 3.6. Raman Imaging with Dye Probe (Rhodamine 6G)

To assess the interaction between chitosan and HPCD, the TFA content was minimized for the Chi:HPCD 2:50 fiber by exposing the fibers to the open atmosphere under adequate ventilation at ambient conditions. This setup favored the volatilization of excess TFA that may influence the sensitivity of the fiber characterization by reducing contributions of high spectral intensity that arise from TFA. The combined use of Raman microimaging along with a suitable dye probe afforded a spectral method that highlighted the chitosan fiber domains that enable the spectral characterization of the regions of interest. Rhodamine 6G was chosen as the dye probe for this study because of the reported favorable binding of chitosan [47] relative to HPCD with the Rhodamine 6G dye [48,49]. Benzene was used as the solvent to avoid the dissolution of fiber components and to maintain the integrity of the electrospun fiber system as it is a poor solvent for such polysaccharide systems (Chi:HPCD). The characteristic Raman signature at 610 and 850 cm$^{-1}$ corresponded to the C–C–C ring *in-plane* bending for Rhodamine 6G [50,51] and the respective ring breathing band for the HPCD glucopyranose unit [44] that was used to construct the 2-D Raman images.

Raman microimaging results in the presence of the Rhodamine 6G dye probe are shown in Figure 5. The Raman spectral image of the HPCD area generated using the Raman signal at 850 cm$^{-1}$ for the fiber is shown in Figure 5A. From this 2-D Raman image, fiber materials with a heterogeneous morphology of beads and fibers concur with the SEM results (Figure 1). The 2-D Raman spectral imaging of the chitosan fraction was generated using the Raman band at 610 cm$^{-1}$, as shown in Figure 5B. It appears that chitosan adopts a *bundle-type* structure in various sample loci. Upon comparing Figure 5A,B, the spectral region for the chitosan domains were not congruently matched with the spectral region of HPCD and may indicate that the composition of the electrospun fiber was heterogeneous in nature. The occurrence of such heterogeneities likely occurs during the electrospinning process. Despite the efforts that were made to remove TFA from the fiber, the persistence of Raman signatures of TFA were noted at ~730 cm$^{-1}$ but with a reduced spectral intensity.

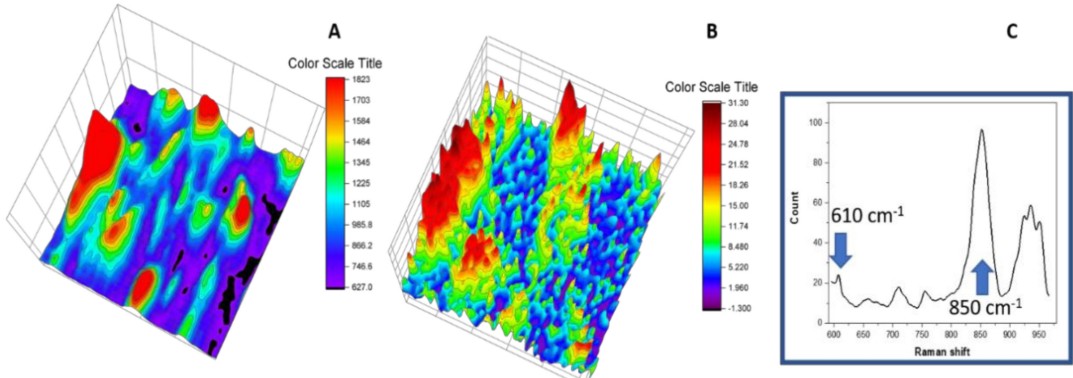

**Figure 5.** Raman imaging results of dried Chi:HPCD 2:50 fiber after soaking with Rhodamine 6G in a benzene solution, where a sample Raman spectrum centered at 790 cm$^{-1}$ was given under static conditions ($\lambda_{ex}$ = 785 nm). The Raman spectral image of the HPCD area (**A**) was constructed by peak integration at 850 cm$^{-1}$. The Raman spectral image of chitosan in area (**B**) was constructed by dividing the spectral intensity for Rhodamine 6G (610 cm$^{-1}$) against that for HPCD (850 cm$^{-1}$) for each respective 1-D spectrum, where a 1-D Raman spectrum is shown in panel C as an example.

Figure 5A,B was combined into one additive 2-D image, as shown in Figure 6 to emphasize different structural domains of the fiber: a chitosan rich area, a chitosan/HPCD mixed area and a HPCD rich area. In the spectrum for the chitosan rich domain (Figure 6A), a Raman signature related to the ring breathing mode of chitosan was noted at 895 cm$^{-1}$. This band had no apparent spectral change when compared against the Raman spectrum of pristine chitosan (Figure 4B). Moreover, the shape of the spectral band at ~850 and ~930 cm$^{-1}$ for all samples (Figure 6A–C) were nearly identical. This may

further suggest that HPCD has no direct intermolecular interactions with the chitosan polysaccharide chain of the fiber.

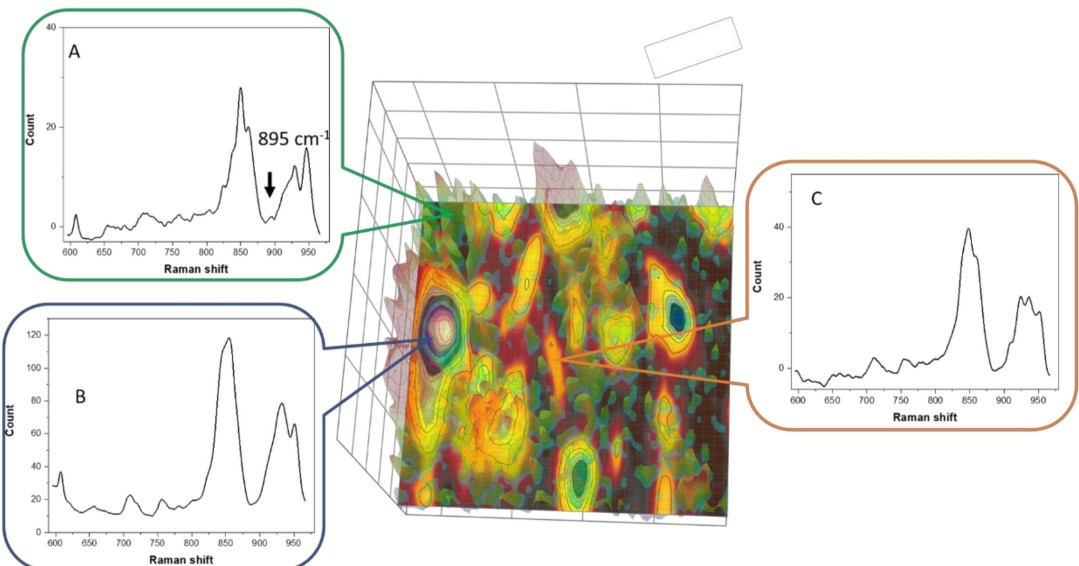

**Figure 6.** A combined Raman image from the spectral data for the chitosan region (Figure 5A) and the HPCD region (Figure 5B). Sample spectrum was shown for different highlighted spectral regions: (**A**) a chitosan rich domain, (**B**) a chitosan/HPCD mixed domain and (**C**) an HPCD rich domain.

### 3.7. Composition of Chi:HPCD Electrospun Fiber and Its Component Interactions

An illustration of compositional change of the Chi:HPCD electrospun fiber over time is shown in Scheme 2. According to the FT-IR and Raman spectral results, TFA existed in the Chi:HPCD electrospun fiber as a third component that underwent slow release from the fiber over time that led to composition and/or structural changes of the fiber. Electrostatic interactions were likely to occur between the –COO$^-$ group from the TFA and NH$_3^+$ group from chitosan under such conditions. A host–guest complex was inferred to occur between the –CF$_3$ group in TFA and HPCD, as described above. This was firstly supported by the disappearance of the C–F band at 1221 and 1183 cm$^{-1}$ according to FT-IR spectra. Secondly, this was confirmed by reduction of the relative intensity of –CF$_3$ signatures in the Raman spectra. Thirdly, further verification was judged by the Raman spectral modifications of the ring breathing region of HPCD for the Chi:HPCD fiber. Thus, the majority of remaining TFA in the fiber may have served as a "connector unit" between chitosan and HPCD, where the –COO$^-$ group of TFA was associated with the NH$_3^+$ group of chitosan via ion–ion interactions, while the –CF$_3$ group of TFA formed an interfacial complex with HPCD. To afford this type of host–guest complex, the wider annular region that contained the secondary hydroxyl groups of HPCD was implicated, since the narrow annular region contained C6-hydroxyl and C6-hydroxypropyl substituents that may have resulted in steric effects. Upon ion–ion binding of TFA with the charged amine sites (–NH$_3^+$) of chitosan, repulsive electrostatic forces between chitosan polymer chains became attenuated. As well as this, the presence of bound TFA onto chitosan with subsequent binding by HPCD may have afforded additional charge screening that led to improved electrospinning performance of the system, as illustrated in Scheme 2. Based on the 2-D Raman microimaging results, compositional heterogeneity of Chi:HPCD electrospun fiber was depicted, where no evidence of a direct or well-defined host–guest interaction between chitosan and HPCD for the fiber system was supported by the Raman spectral results. In turn, this was consistent with the likely formation of a "facial complex", as reported for the case of a β-CD/halothane complex with a reported 1:1 binding constant ca. $10^2$ M$^{-1}$ in aqueous media [32].

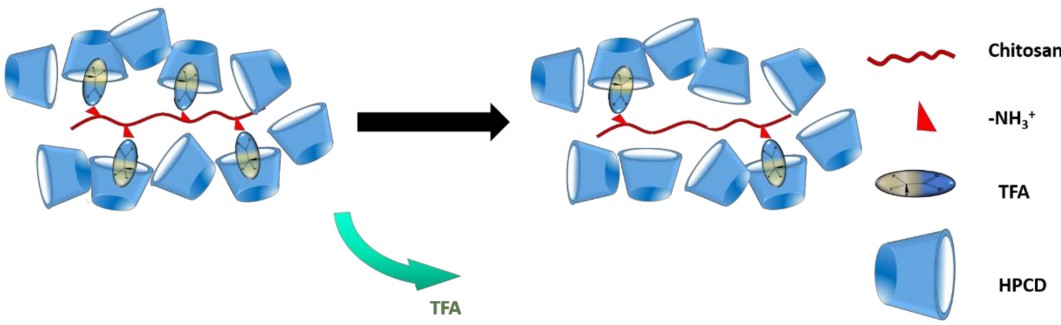

**Scheme 2.** An illustrative view of the compositional change of a Chi:HPCD electrospun fiber over time, where the green arrow shows incremental temporal loss of trifluoroacetic acid (TFA).

## 4. Conclusions

Chi:HPCD 2:20 and 2:50 fiber were produced via electrospinning of a mixture of HPCD and chitosan using TFA as a solvent. The composition of Chi:HPCD fibers was determined using thermal analysis and complementary spectral methods. TFA was found to be a constituent in the Chi:HPCD fiber assembly. The heterogeneous morphology and composition of this electrospun fiber was revealed using SEM and Raman imaging, along with a dye-based probe method [34]. Interactions among the components were also characterized using complementary methods, such as IR/Raman spectroscopy. TFA appears to have multifunctional properties as a solvent during the electrospinning process, but also protonates chitosan and stabilizes host–guest interactions with HPCD. Hence, the presence of lateral binding sites along the chitosan backbone (*cf.* Scheme 2) due to electrostatically bound TFA are found to play a crucial role in the effective dispersion of chitosan and the reduction of repulsive forces during electrospinning between chitosan polymer chains that favor fiber formation. There was no clear evidence of direct interactions between the glucosamine moiety of chitosan and HPCD in the solid-state according to Raman imaging, along with the dye probe method. The combined complementary results herein account for the fiber formation process as the role of weak host–guest supramolecular interactions that arise from the formation of a facial complex, which have been independently reported [32]. In turn, we envisage that such controlled-release supramolecular disassembly will contribute to the development of "smart coatings" that may utilize diverse types of chitosan polyelectrolyte complexes.

**Supplementary Materials:** The following are available online at http://www.mdpi.com/2079-6439/7/5/48/s1, Figure S1: FT-IR spectrum (**A**) and a DTG plot (**B**) of pristine chitosan; Figure S2: $^1$H NMR spectra of pure HPCD (**A**) and Chi:HPCD 2:20 fiber (**B**) prepared in 1% (*w/w*) tetrahydrofuran (THF)/DMSO-$d_6$ solution for HPCD content determination; Figure S3: Solid-state $^{13}$C CP-TOSS NMR spectra of HPCD (**A**), Chitosan (**B**) and Chi:HPCD (**C**) 2:50 fiber (from top to bottom, respectively).

**Author Contributions:** Conceptualization, C.X. and L.D.W.; methodology, C.X.; software, C.X.; validation, C.X. and L.D.W.; formal analysis, C.X.; investigation, C.X.; resources, L.D.W.; data curation, C.X.; writing—original draft preparation, C.X.; writing—review and editing, C.X. and L.D.W.; visualization, C.X.; supervision, L.D.W.; project administration, L.D.W.; funding acquisition, L.D.W.

**Funding:** This research was funded by the Government of Canada through the Natural Sciences and Engineering Research Council (NSERC), Discovery Grant Number: RGPIN 2016-06197.

**Acknowledgments:** The Natural Sciences and Engineering Research Council (Discovery Grant Number: RGPIN 2016-06197) and the University of Saskatchewan are gratefully acknowledged for support of this research.

**Conflicts of Interest:** The authors declare no conflict of interest.

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
