# Peer review of "A Spectroscopic Study of Solid-Phase Chitosan/Cyclodextrin-Based Electrospun Fibers"

_fibers, doi:10.3390/fib7050048_

Round 1
Reviewer 1 Report
1. Authors could give more explanation on why the usage of chitosan/HPCD nanofiber materials are deserving to be given more attention and further investigation.
2. Basically, this manuscript could be seen as extensive research of Burns et al. (reference No. 31). The description of the difference between these two researches could be given more at the Introduction and Conclusion sections in this manuscript.
3. TFA is a volatile and harmful solvent to people and equipment, the remaining amount of TFA in the as-spun fiber will influent the testing results, therefore
a. How to set up the testing time from the ending of electrospinning to starting a different spectroscopic exam to be the same?
b. How to sure the TFA in the as-spun fibers is keeping at the same amount.
Author Response
Author Response to Reviewer comments on MS ID: fibers-487185
Reviewer #1:
1. Authors could give more explanation on why the usage of chitosan/HPCD nanofiber materials are deserving to be given more attention and further investigation.
Author response: Chitosan/HPCD nanofibrous materials may have significant impacts on applications in medical devices, food, and the cosmetics industry (i.e., wounding dressing, food packaging/preservative, coating materials, etc.) due to the unique inclusion complexation properties of chitosan and HPCD for controlled-release applications. For example, highly volatile compounds such as essential oils or water insoluble drugs can be stabilized and used for controlled-release applications or to improve water solubility, viscosity, etc. Meanwhile, the design of chitosan/HPCD formulations is noteworthy of further investigation because HPCD has a dual role since it is working not only as an additive polymer to improve the electrospinning performance, since it also serves as a carrier matrix for compounds, as mentioned above.
Celebioglu, A.; Yildiz, Z.I.; Uyar, T. Electrospun nanofibers from cyclodextrin inclusion complexes with cineole and p-cymene: enhanced water solubility and thermal stability. Int. J. Food Sci. Technol. 2018, 53, 112–120.
Samprasit, W.; Akkaramongkolporn, P.; Kaomongkolgit, R.; Opanasopit, P. Cyclodextrin-based oral dissolving films formulation of taste-masked meloxicam. Pharm. Dev. Technol. 2018, 23, 530–539.
2. Basically, this manuscript could be seen as extensive research of Burns et al. (reference No. 31). The description of the difference between these two researches could be given more at the Introduction and Conclusion sections in this manuscript.
Author response: Burns et al. (see Ref. 31 in manuscript) attempted to understand the mechanism that improves the electrospun fiber formation of chitosan using mainly solution-based characterization method (solution 2D NMR). In their paper, they inferred that the complex was formed between chitosan and HPCD which results in the better electrospun fiber. They also discussed solution rheology for different concentrations of acetic acid (AA) and its effect on the fiber morphology. However, they did not study TFA related solution sample due to its corrosive nature and there was little discussion of the role of acetic acid AA or TFA on the fiber formation process. The key evidence (2D NMR) presented by Burns et al. to provide evidence of the complex formation between chitosan and HPCD was not unequivocal.
By contrast, the role of TFA on the electrospinning performance was noted in our independent work and this led us to carry out further research to provide molecular-level insight on the fibre formation mechanism reported herein. In our study, we focused on various solid-state characterization methods (FT-IR, Raman, and Raman imaging) that are complementary to the solution-based study in the initial report by Burns et al. The results highlighted the multifunctional role of TFA as a solvent, proton donor, and electrostatically bound pendant group to chitosan, where TFA favours the formation of a ternary complex via supramolecular host-guest interactions. The role of HPCD appears to stabilize the electrostatic complex between TFA and the protonated amine sites of chitosan, along with charge shielding effects to minimize repulsive interactions between ionic charges along the chitosan backbone.
3. TFA is a volatile and harmful solvent to people and equipment, the remaining amount of TFA in the as-spun fiber will influent the testing results, therefore
a. How to set up the testing time from the ending of electrospinning to starting a different spectroscopic exam to be the same?
b. How to sure the TFA in the as-spun fibers is keeping at the same amount.
Author response:
While we agree with the comments by the Reviewer, TFA was used as the solvent system herein to address the role of this solvent in the fiber formation process. It is possible that an alternative solvent system such as supercritical CO2 or an ionic liquid may be used with stoichiometric amounts of TFA or acetic acid to afford charge neutralization, however; this is beyond the scope of the present study and will be considered in future work. In a fabrication process where excess TFA is used, controlled evaporation of fibers under vacuum or controlled heating can be carried out to ensure quality control of resulting fiber materials instead of longer drying at ambient conditions.
An assumption made herein is that bound TFA in the sample can undergo evaporation from the as-spun fiber at a fixed rate under ambient conditions. In this study, different sample preparations were allowed to undergo drying overnight for 12 hours after the electrospinning process to ensure that the loss of TFA was similar across all the samples. In the case of ionized TFA that is bound via ionic interactions with chitosan, the presence of moisture is anticipated to affect the loss of TFA. Therefore, long-term storage of fibers can be carried out under controlled humidity to afford fibers with uniform sample composition.
The authors wish to thank Reviewer #1 for the constructive and insightful comments provided, along with the opportunity to improve the quality of this manuscript submission.

Reviewer 2 Report
The manuscript “A Spectroscopic Study of Solid-Phase Chitosan/Cyclodextrin-Based Electrospun Fibers” reported chitosan/6-O-hydroxypropyl β-cyclodextrin (Chi/HPCD) fibers with different ratio assembled via an electrospinning method using TFA as a solvent. The authors claimed that the results highlight the multifunctional role of TFA as a solvent, proton donor, and electrostatically bound pendant group to chitosan that results in the formation of a ternary complex via supramolecular host-guest interactions. The authors have provided solid data to back up the conclusions in most cases. However, when reading the manuscript many questions arise, therefore some complementary information and revision should be taken into account before being published in “Fibers”.
1. The abbreviations should be defined first before using, such as “TFA” in Abstract. Also, there is an incorrect word “hydroxypropyl- -CD” in Abstract. Please check the manuscript carefully before submission.
2. In Introduction, “To address this goal, preparations of Chi:HPCD fibers at variable ratios via electrospinning in nonaqueous media at variable conditions.” is not a sentence.
3. In section 3.3, the spectra of Chi should also be given as the baseline to prove the as-mentioned TFA peaks are not coming from Chi.
4. In section 3.4, please provide the DTG plots of Chi and 2:20 fibers for comparing. The small peak at ~260 °C is needed to be proven not from Chi. Besides, what ratio was used for the physical mixture in Figure 3b?
5. In Figure 4, a strange peak around 2300 can be only found from the spectra of C and F, which should be discussed further.
6. In section 3.6, the authors applied efforts to remove TFA from the fibers. Have vacuum dry or rotary evaporation been dried? TFA should be easy to remove thoroughly.
7. In Figure 5, the caption of C was missing.
8. In Conclusion, usually references are not supposed to be shown since it should be summarized from the manuscript. It is especially not appropriate to cite the authors own papers. If the same conclusion can be obtained from the previous research, it is not necessary to publish this manuscript. Please reorganize the Conclusion and remove the references.
9. The English writing in manuscript needs to be checked very carefully before submission since the meaning of many sentences cannot be understood by the improper use of English or ambiguous description.
Author Response
Author Response to Reviewer comments on MS ID: fibers-487185
Reviewer #2
1. The abbreviations should be defined first before using, such as “TFA” in Abstract. Also, there is an incorrect word “hydroxypropyl- -CD” in Abstract. Please check the manuscript carefully before submission.
Author response:
The corresponding corrections were made according to the reviewer suggestion.
2. In Introduction, “To address this goal, preparations of Chi:HPCD fibers at variable ratios via electrospinning in nonaqueous media at variable conditions.” is not a sentence.
Author response:
Corrections were made according to the reviewer comment.
3. In section 3.3, the spectra of Chi should also be given as the baseline to prove the as-mentioned TFA peaks are not coming from Chi.
Author response: As recommended by the reviewer, IR spectrum of chitosan was provided in the supplementary information (Figure S1A).
4. In section 3.4, please provide the DTG plots of Chi and 2:20 fibers for comparing. The small peak at ~260 °C is needed to be proven not from Chi. Besides, what ratio was used for the physical mixture in Figure 3b?
Author response:
The DTG of chitosan is shown in the supplementary information (Figure S1B).
TGA was used to investigate the composition of as-spun fiber. The DTG plot of as-spun chi:HPCD 2:20 was not provided here because the free TFA, whose existence was directly supported by IR, could results in damage to the instrument. On the other hand, dried Chi:HPCD 2:20 sample was different from as-spun samples and cannot be used for comparison.
The mass ratio between Chitosan and HPCD of the physical mixture presented herein was 2:50.
5. In Figure 4, a strange peak around 2300 can be only found from the spectra of C and F, which should be discussed further.
Author response:
Because the Raman band near 2300 cm-1 was not observed consistently in Chi:HPCD 2:50 samples, this peak most likely is attributed to an artefact such as a cosmic ray, which is known to occur in Raman spectroscopy.
6. In section 3.6, the authors applied efforts to remove TFA from the fibers. Have vacuum dry or rotary evaporation been dried? TFA should be easy to remove thoroughly.
Author response:
The sample was dried under open ventilation at ambient conditions. Free TFA was removed from the fiber but its ionized form (trifluoroacetate ion) remained in the fiber. In the absence of a washing step, the TFA anion is strongly bound to the protonated amine sites of chitosan, as anticipated.
7. In Figure 5, the caption of C was missing.
Author response: The caption was revised accordingly.
8. In Conclusion, usually references are not supposed to be shown since it should be summarized from the manuscript. It is especially not appropriate to cite the authors own papers. If the same conclusion can be obtained from the previous research, it is not necessary to publish this manuscript. Please reorganize the Conclusion and remove the references.
Author response: While the authors appreciate the comment by the reviewer, the citation from the study by Wilson and Verrall is a completely independent study with no overlap to the current study. It is our opinion that the structure of such β-CD/halothane complexes lends a significant contribution to the interpretation presented in the present work. Thus, we feel it is appropriate to make this connection and cite the report accordingly for interested readers to obtain a more complete description of the interpretation presented herein.
9. The English writing in manuscript needs to be checked very carefully before submission since the meaning of many sentences cannot be understood by the improper use of English or ambiguous description.
Author response: The entire manuscript was further revised for language editing, syntax, and clarity throughout to meet the high standards of this journal.
The authors wish to thank Reviewer #2 for the constructive and insightful comments provided, along with the opportunity to improve the quality of this manuscript submission.

Round 2
Reviewer 2 Report
All the questions and comments raised by the reviewers were legitimately explained and revised. The accuracy and detail of the manuscript were also improved further after revision. Therefore, the revised manuscript is now suitable publication in "Fibers".